# Initial Glaucoma Medication in the Hypertensive Phase Following Ahmed Valve Implantation: A Comparison of Results Achieved Using Aqueous Suppressants and Prostaglandin Analogs

**DOI:** 10.3390/jcm9020416

**Published:** 2020-02-03

**Authors:** Jiyun Lee, Chan Kee Park, Kyoung In Jung

**Affiliations:** Department of Ophthalmology, Seoul St. Mary’s Hospital, College of Medicine, The Catholic University of Korea, Seoul 06591, Korea

**Keywords:** Ahmed glaucoma valve, aqueous suppressant, prostaglandin analogs, hypertensive phase, intraocular pressure, glaucoma

## Abstract

Background: To compare the effects of aqueous suppressants (AS) and prostaglandin (PG) analogs during the hypertensive phase on intraocular pressure (IOP) and surgical outcomes. Methods: In this retrospective, observational study, 66 eyes (66 patients) with Ahmed glaucoma valve (AGV) implantation were included. As evaluation items, IOP changes, number of postoperative medications, the surgical success rate, and postoperative complications were examined. Complete success was defined as IOP between 6 and 21 mmHg without medications, while qualified success was with a maximum of four medications. Results: The short-term IOP reduction following initial medication was 9.3 mmHg for AS and 4.4 mmHg for PG analogs (*p* = 0.016). More postoperative medications were used in PG than in AS from postoperative 3 months to 3 years (all *p* < 0.05). The qualified success rate with the initial medication was higher in AS than in PG (67.5% vs. 42.3% at 1 year, 80.6% vs. 37.5% at 2 years, 80.0% vs. 35.0% at 3 years, all *p* < 0.05). Conclusions: Association between AS used as the first medications during the hypertensive phase and better IOP control and a higher success rate was observed. The type of the initial glaucoma medication after AGV implantation could affect short- and long-term surgical outcomes.

## 1. Introduction

Glaucoma drainage devices (GDD) are becoming more commonly used after a failed trabeculectomy or in cases with a high risk of trabeculectomy failure [1]. Although GDDs offer better intraocular pressure (IOP) control and higher surgical success rates, the hypertensive phase (HP), typically defined as an IOP of more than 21 mmHg, can occur at one to three months postoperatively in general [2,3]. According to Nouri-Mahdavi et al. [3], 56% to 82% of patients with Ahmed glaucoma valve (AGV) implantation underwent a HP, and, among 72% the HP did not resolve. Consequently, management of the HP is critical in terms of preventing further damage to the optic nerve and preserving the rest of the visual field. Therefore, recently, clinical studies of initiation of the aqueous suppressant (AS) before onset of the HP have been investigated and the results have shown that early AS treatment can improve IOP reduction, the frequencies of the HP, and the consequent success rate after AGV implantation [4,5]. According to these findings, it seems beneficial for patients’ disease course to start medication in advance to the onset of the HP. However, in an office, it is not easy and simple to ask patients to resume glaucoma medication before the development of the HP. However, needless to say it has been an inevitable decision for clinicians to start medical therapy at the onset of the HP [2].

In our previous animal experimental study [6], we found that postoperative IOP and the fibrosis of blebs analyzed by immunohistochemical staining of α–smooth muscle actin (SMA) or histochemical staining of collagen was significantly lower in the AS group than in either the prostaglandin (PG) analogs group or the control group. In addition, in tenon tissues, Interleukin (IL) -2, one of the inflammatory cytokines was higher in PG analogs group than either the AS or the control group. 

As for PG analogs, one of the most commonly used medications, its mechanism of IOP reduction by PG analogs is that they induce relaxation of the ciliary muscle and modify its extracellular matrix [7]. Also, PG is involved in the course of inflammatory disease [8]. Liu et al. found that collagen gel contraction was induced by latanoprost [9]. Considering these findings, the use of either aqueous suppressants or PG analogs might affect the modulation of the wound healing process. 

We, therefore, hypothesized that the initial use of AS or PG analogs might result in different clinical outcomes. Also, there have been no studies published to date comparing the effect of AS and PG analogs as an initial medication during the HP after AGV implantation in human clinical settings. We analyzed the effects of AS and PG analogs given during the HP on IOP and surgical outcomes after implantation of AGV.

## 2. Experimental Section

This retrospective observational study was approved by the Institutional Review Board of the Catholic University of Korea in Seoul, Korea (IRB approval number KC19RESI0658). The study design followed the tenets of the Declaration of Helsinki for biomedical research. Patients with glaucoma who underwent AGV implantation from March 2012 through June 2013 and visited the outpatient clinic at the Department of Ophthalmology, Seoul St. Mary’s Hospital, for postoperative follow-up for at least one year were included. Patients with a follow-up period of less than one year or who were followed up without medications due to no presence of the HP were excluded. Among the patients who were followed up for more than a year, if postoperative complications such as valve exposure or endophthalmitis, bleb infection or tube obstruction, and additional glaucoma surgery due to inadequate IOP control took place, data before the event were included and used for analysis. 

All operations were performed by a single surgeon (C. K. P.) using model FP7 AGVs (New World Medical Inc., Rancho Cucamonga, CA, USA) with anterior chamber tube placement. First, a fornix-based incision was made through the conjunctiva and Tenon’s capsule in the superior temporal quadrant, followed by radial relaxing incisions on one side of the conjunctival flap to improve surgical exposure. The plate was inserted under the conjunctiva and Tenon’s capsule and secured to the sclera with non-absorbable sutures located 8 mm posterior to the limbus. Priming of the valve was performed by irrigation with balanced salt solution through the tube, ensuring patency before insertion. The tube was cut to the appropriate length with a beveled edge so that it extended approximately 2 to 3 mm into the anterior chamber. Next, a paracentesis was created with a 23-gauge needle from 1 to 2 mm posterior to the limbus. Viscoelastic was injected into the anterior chamber through the same needle, and the tube was inserted into the anterior chamber. The tube was then loosely secured to the sclera with a 10-0 nylon suture. Partial ligation of the tube was performed with 9-0 nylon sutures. The extraocular portion of the tube was covered with a 4 × 4 mm^2^ human donor sclera that was sutured with 10-0 nylon. The conjunctiva was reapproximated with 8-0 vicryl sutures (Ethicon Inc., Somerville, NJ, USA). Topical antibiotics and steroids were used after surgery, and the releasable suture was removed during the first postoperative week in all patients.

Postoperative follow-up visits were scheduled on Days 1, 2, and 7 and Months 1, 2, 3, and 6 after the operation and every year thereafter. The hypertensive phase was defined as when IOP exceeded 21 mmHg during the postoperative first three months. Therefore, when the IOP was over 21 mmHg during the first three months, aqueous suppressants or PG analogs were initiated to overcome the hypertensive phase. Dorzolamide hydrochloride plus timolol maleate fixed-combination drops and brinzolamide plus timolol maleate fixed combination were used for aqueous suppressant treatment, while latanoprost, bimatoprost, or travoprost were used when PG analog therapy was required. The aqueous suppressants group received their medication twice daily, while the PG group received their medication once daily. The initial medication during the hypertensive phase was selected at the discretion of the operating surgeon depending on glaucoma medication of the fellow eye or history of previous allergic reaction to glaucoma eye drops or other relevant factors. If the IOP was not properly controlled with a single medication, supplemental medical therapy could be added. Surgical therapy could also be pursued to protect the patients from unnecessary exposure to high IOP when the IOP was not controlled with maximum medical treatment. 

We reviewed each patient’s chart and collected clinical data including age, sex, diabetes, or hypertension as well as a comprehensive ophthalmic examination results such as slit-lamp biomicroscopy, gonioscopy, central corneal thickness measurements using ultrasonic pachymetry (SP-3000; Tomey Corp., Tokyo, Japan), dilated fundus observation, mean deviation, and pattern standard deviation on a Humphry visual field analyzer (Carl Zeiss Meditec, Dublin, CA, USA) (24-2 Swedish interactive Threshold Algorithm Standard). The type of glaucoma, history of previous intraocular surgeries, lens status, and preoperative and postoperative numbers of medications were investigated as well. In terms of postoperative numbers of medications, the numbers were counted as components not as bottles. The primary outcome measures were IOP, number of postoperative medications, and success rate. Complete success was defined as an IOP of between 6 and 21 mmHg without any medication, while cases that qualified as successful fell within the same IOP range but included a maximum of four glaucoma medications. The remaining cases were classified into the failed category. Other outcomes measures included postoperative complications such as choroidal detachment, shallow anterior chamber, hypotony, decompressive retinopathy, disc hemorrhage, tube malposition, tube exposure, and malignant glaucoma and vascularity changes around the AGV including blebs and plate margins before and after the commencement of medications. 

Statistical analyses were performed using the Statistical Package for the Social Sciences version 17.0 software program (IBM Corp., Armonk, NY, USA). All data are represented as means ± standard deviations and frequency values (i.e., percentages). To compare the differences between the groups, we used Student’s *t*-test, the chi-squared test, and Fisher’s exact test. 

Logistic regression analysis was used to identify risk factors predictive of surgical failure with the supplemental use of medication after AGV implantation. Success rates over time in the two groups were compared by the Kaplan–Meier survival log-rank test. A *p*-value of 0.05 or less was considered to be statistically significant in our analyses.

## 3. Results

### 3.1. Baseline Characteristics

Initially we examined 86 eyes of 86 patients; however, 17 eyes among subjects who did not experience the HP were excluded. Additionally, two eyes who commenced medical treatment with brimodinine and one eye using a PG analog and AS concurrently were excluded as well. Exclusion due to follow-up less one year had not found.

Of the remaining 66 eyes among patients with a mean age of 54.9 ± 16.2 years, 40 eyes received AS and 26 eyes received PG analogs. After 1:1 propensity score matching, 22 patients were selected from 26 patients with PG analogs. The total mean follow-up period length was 32.0 ± 8.5 months, with no significant difference found between the two groups (*p* = 0.634). Table 1 summarizes the demographic characteristics of the study groups without any statistically significant differences, including preoperative visual field status before and after propensity score matching (all *p* > 0.05).

### 3.2. Changes of Intraocular Pressure (IOP) and Numbers of Medications before and after Ahmed Glaucoma Valve (AGV) Implantation

IOP and the numbers of medications used before and after AGV implantation are shown in Figure 1. There were no significant differences in preoperative IOP and the number of medications used preoperatively (All *p* > 0.005). After AGV implantation, we found that the AS group used a significantly fewer mean number of medications than did the PG analogs group from three months to three years after surgery (All *p* < 0.05). After propensity-score matching, the mean number of medications used was significantly greater in the PG analog group than the AS group at three months, two years and three years (All *p* < 0.05). Also, IOP at three years post operation was significantly lower in the AS group than in the PG analogs group (All *p* < 0.05 before and after propensity score matching). In addition, short-term IOP changes defined as differences between IOP measured at the onset of the HP and IOP measured at the immediate next visit after the initiation of the medication were significantly higher in the AS group than in the PG analogs group (All *p* < 0.05; Figure 2). 

### 3.3. Postoperaitve Complications 

No significant postoperative complications differences before and after propensity score matching were observed (Table 2). 

### 3.4. Success Rate Analysis Between Aqueous Suppressants (AS) and Prostaglandin (PG) Analogs Groups 

For further analysis of the success rate, different approaches were attempted, one considering complete success and the other considering qualified success. In terms of complete success, there were no significant differences between the two groups during the total follow-up period before and after propensity-score matching (all *p* > 0.05). The other approach, which concerned the qualified success rates at one year, two years, and three years with using two different criteria (one with only one initial medication, either one fixed-combination AS or one PG analogue and the other with a maximum of four medications), was also analyzed (Table 3). With only one initial medication, the AS group showed a significantly superior success rate to PG analogs at all three years after surgery (*p* = 0.043, *p* = 0.001 and *p* = 0.001, respectively), but, after propensity-score matching, the success rate only at two and three years reached statistical significance (*p* = 0.028 and *p* = 0.010, respectively). With a maximum of four medications, the success rate of the AS group before propensity score matching was significantly greater than that of the PG analogs group at three years after surgery (*p* = 0.039), after propensity-score matching, meanwhile, no significance was found during total follow-up period. 

In a multivariate logistic regression analysis, only the postoperative initial use of PG analogs medication turned out to be a risk factor for the failure of AGV implantation (*p* < 0.001 before propensity score matching, *p* = 0.009 after propensity score matching; Table 4). 

Kaplan–Meier survival analysis was adopted to compare surgical success rates over time between the two groups, and two different criteria of number of medications were applied. In terms of only one initial medication, which stands for one fixed-combination AS or one PG analogue, the cumulative probability of the success was 82.9% in the AS group and 35% in the PG analogs group at three years postoperatively (*p* < 0.001, log-rank test; Figure 3A). With a tolerance of maximum three medications, the cumulative probability of the success was 97.1% in the AS group and 55% in the PG analogs group at three years postoperatively (*p* < 0.001, log-rank test; Figure 3C). After propensity score matching, the cumulative probability with tolerance of maximum three medications was 90% and 55.6%, respectively (*p* = 0.004, log-rank test; Figure 3D) and that of with only one initial medication was 80% and 38.9%, respectively (*p* = 0.014, log-rank test; Figure 3B). 

Representative images of each group are shown in Figure 4. A 43-year-old female patient who used AS during the HP exhibited decreased vascularity around the bleb when compared with before beginning glaucoma medication, while a 69-year-old male patient who started PG analog treatment presented relatively increased vascularity. 

## 4. Discussion

In this study, we found that starting IOP control with AS during the HP after AGV implantation resulted in a greater IOP reduction than using PG analogs in the short-term period.

Numerous studies have shown that fixed combinations of AS and PG analogs are comparably effective at lowering IOP in glaucoma surgery-naïve eyes [10,11,12,13]. However, in eyes with AGV implantation, the amount of IOP reduction was significantly greater in the AS group than the PG analogs group, although both medications lowered IOP during the HP. We presume that the better IOP control achieved with initial AS might be related to the dissimilar effects of AS or PG analogs on wound healing modulation following AGV implantation, not the direct IOP-lowering effects of them.

It is already known that the HP after AGV implantation is part of the natural course of wound healing to form a fibrotic encapsulated bleb [2,3,14]. During the HP, pre-existing inflammatory cytokines related to fibrotic responses in the glaucomatous aqueous humor [15,16,17,18,19] are secreted more strongly. Freedman et al. [20] found that the amounts of transforming growth factor β (TGF-β) and PGE2 were higher in the aqueous humor of patients with a HP after Molteno implantation than in those of patients with a trabeculectomy or the control group. The authors’ following study revealed that cytokines including TGF-β2, interleukin (IL)-6 were highest in the Molteno bleb group with a HP [21]. These increased cytokines might aggravate fibrous bleb formation [22]. In addition, during HP, the mechanical application due to high IOP in the bleb might promote fibroblasts to be activated and transformed into myofibroblasts [23,24]. Considering the changes in the environments following AGV implantation, AS during the HP might have acted as antifibrotic agents by not only diminishing the secretion of aqueous humor and its inflammatory cytokines but also decreasing the aqueous pressure in the bleb.

Our findings were in accordance with the precedent clinical studies of the initiation of AS before the onset of the HP [4,5]. The difference between our study and their study was the time of initiation of glaucoma medication. They began glaucoma eye drops before the development of the HP [4,5], meanwhile, we started glaucoma medication after the development of the HP. When AS was used as an initial IOP-lowering medication, even after the development of the HP, they ensured better postoperative outcomes than the PG analogs. Therefore, AS could be considered as a reasonable initial option to minimize the deleterious effect of the HP and as a means to promote favorable surgical outcomes after AGV implantation.

About PG analogs, the potency of the IOP-lowering effect was relatively less efficient than that of AS during the HP after AGV implantation. PG analogs have been shown to be related to inflammatory reactions [25] and there was a report that PGF2α was involved in acute and chronic inflammatory disease [8]. Kwon et al. [26] reported that the VEGF (Vascular Endothelial Growth Factor) concentration was increased in the aqueous humor of glaucoma patients with the use of PG analogs. Due to the increased possibility of enhancing bleb fibrosis by PG-induced inflammation, consequently, use of PG analogs as initial medications might exhibit inferior outcomes when compared with AS. According to some studies [27,28], the fixed combination of AS showed better IOP controls in surgery-naïve eyes than PG analogs, although most studies have shown that fixed combinations of AS and PG analogs are comparably effective at lowering IOP in glaucoma surgery-naïve eyes. Therefore, we could not exclude the possibility that the inferior potency of PG analogs to AS might have been due to fewer classes of medication comprising PG analog than the fixed combination of AS. In addition, Formation of a new outflow channel by the Ahmed valve implant might diminish the ability of PG_mediated uveoscleral outflow pathway to function. That might contribute to less IOP-lowering efficacy of PG analogs after Ahmed valve implantation.

There are some limitations in this study. First, the patients were not assigned on a random basis, so there might be an issue of selection bias. Although a propensity score-matching analysis and multivariate logistic regression analysis were applied, further study should involve a prospective, randomized clinical trial design. Second, a control group without the HP was absent because we focused on the comparison of postoperative outcomes between initial AS and PG treatments in patients with an occurrence of the HP. Third, a statistical assessment of morphological bleb status was not carried out. In terms of evaluating the success of AGV implantation, bleb status as well as IOP change are important in general. Therefore, we tried to look into the bleb morphology, but 22 patients (55%) in the AS group and 14 patients (54%) in the PG analogs group did not have appropriate photos available for review. However, even with these limitations, it represents the first attempt, to our knowledge, to evaluate the effects of AS in comparison with PG analogs during the HP after AGV implantation in the human clinical setting.

## 5. Conclusions

AS showed better IOP control during the HP and a higher surgical success rate after AGV implantation than PG analogs. However, it is not certain that initial treatment with AS might still be superior to PG analogs in the long run when the wound-healing process after AGV implantation is almost completed. Therefore, our findings could be applicable when we resume treatment during the HP or early phase after AGV implantation surgery.

## Figures and Tables

**Figure 1 jcm-09-00416-f001:**
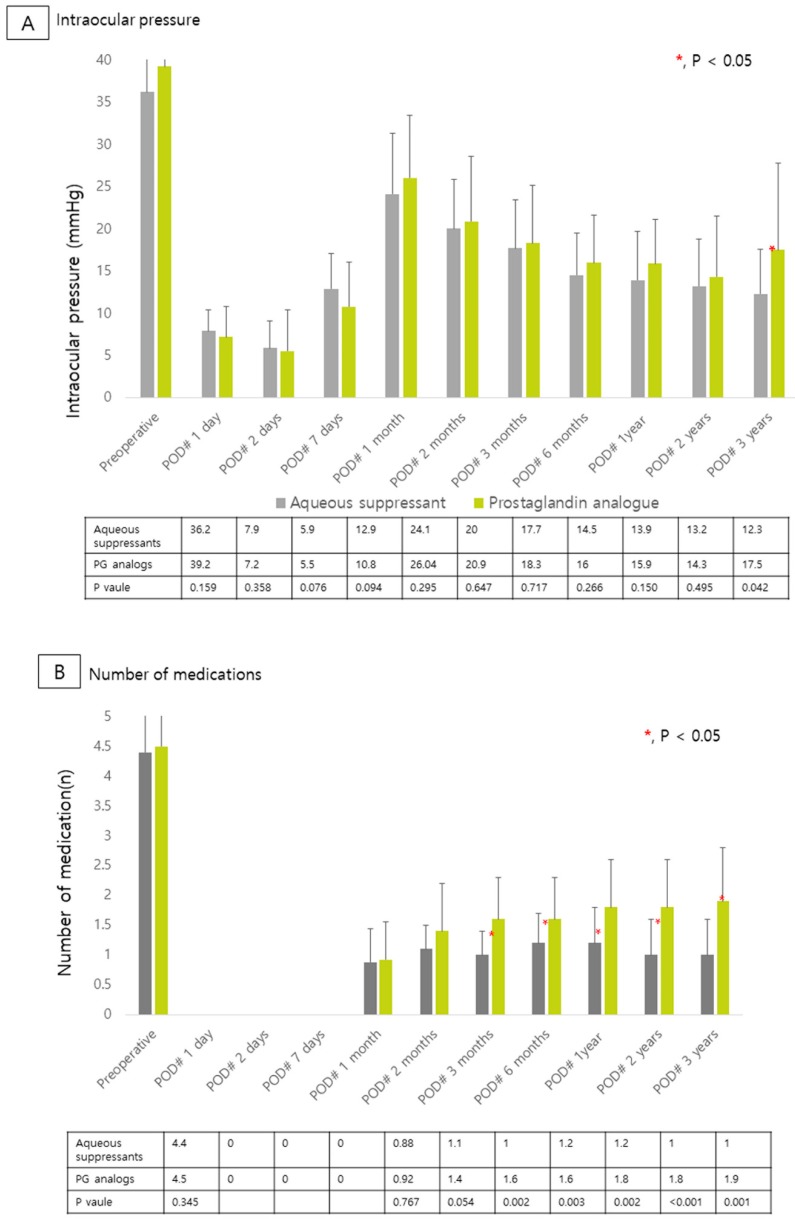
Postoperative intraocular pressure (IOP) and number of medications during follow-up changes in IOP (**A**), and number of medications (**B**) before and after the Ahmed valve implantatio. POD, postoperative day.

**Figure 2 jcm-09-00416-f002:**
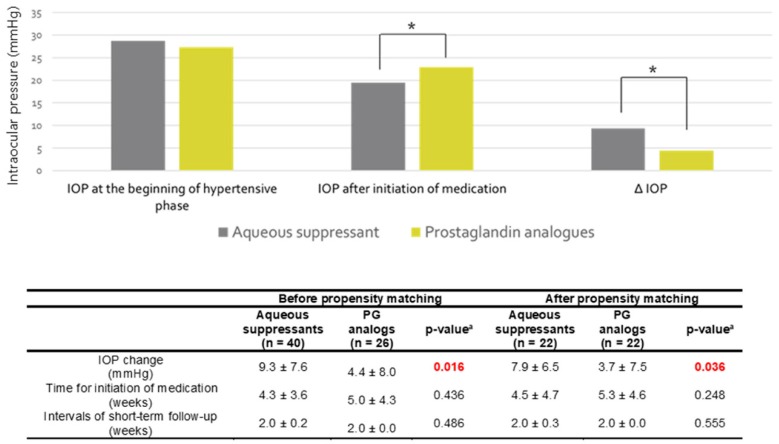
Short-term intraocular pressures (IOP) changes after the initiation of the glaucoma medications. ^a^ Student’s *t*-test Red font indicates significant *p* values (*p* < 0.05), * *p* < 0.05.

**Figure 3 jcm-09-00416-f003:**
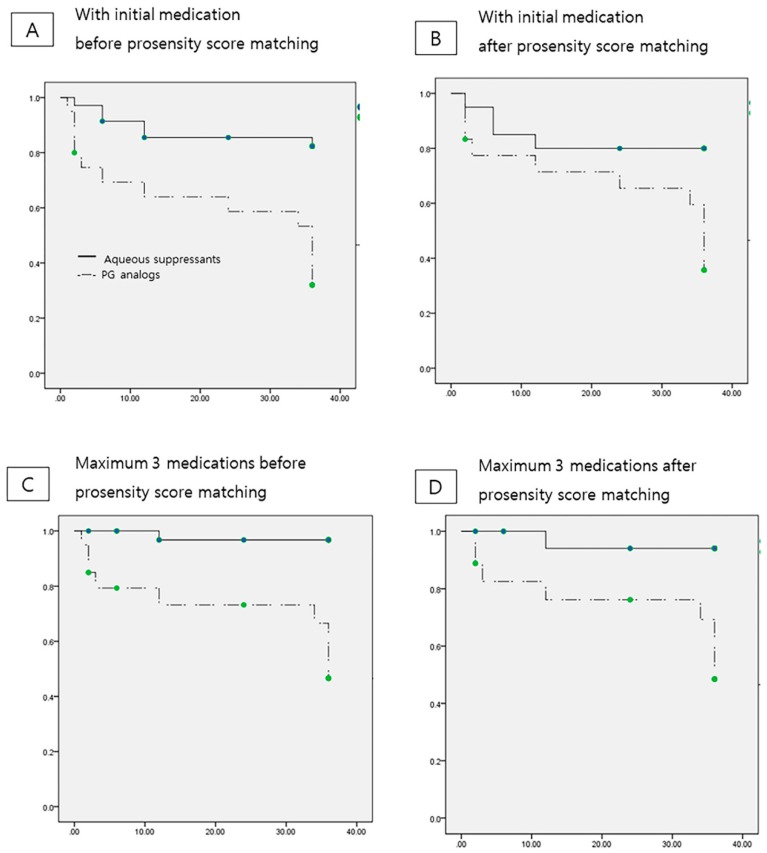
Kaplan–Meier analysis of the cumulative probability of qualified success over time considering initial glaucoma medication in the hypertensive phase after AGV implantation before and after propensity score matching regardless of number of medications used, the cumulative probability of qualified success was higher in the aqueous suppressants group than in the prostaglandin analogs group. With one initial medication, before propensity score matching analysis (**A**), the analysis revealed 82.9% for aqueous suppressants group, 35% for prostaglandin analog group (*p* < 0.001), and after propensity score analysis (**B**), 80% for aqueous suppressants group, 38.9% for prostaglandin analog group (*p* = 0.014). With maximum 3 medications, 97.1% for aqueous suppressants group, 55% for prostaglandin analog group (*p* < 0.001) before propensity score matching analysis (**C**), and 90% for aqueous suppressants group, 55.6% for prostaglandin analog group after propensity score matching analysis (**D**) (*p* = 0.004).

**Figure 4 jcm-09-00416-f004:**
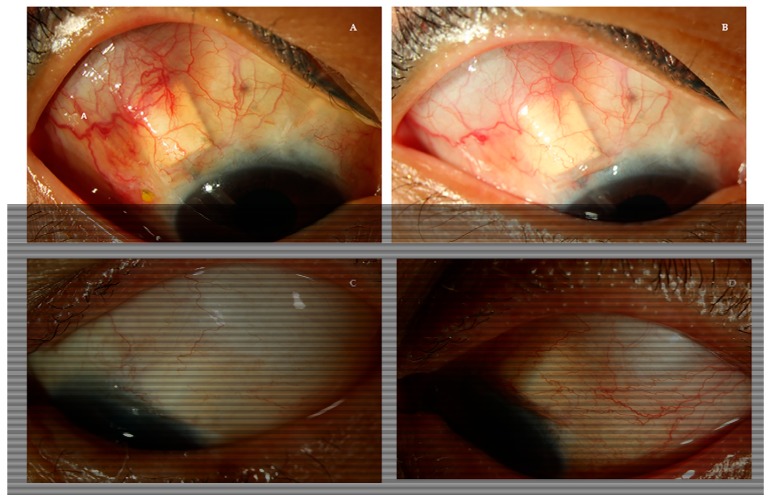
Photographs of vascularity changes around the AGV including the bleb and the plate margin in the AS group and PG analogs group in a short -term period. (**A**,**C**) were taken at the onset of the hypertensive phase and (**B**,**D**) were taken two to four weeks after the initiation of the medication. (**A**,**B**) A 43-year-old female with uveitic glaucoma after AGV implantation started AS (timolol–dorzolamide fixed combination) and the vascularity around AGV decreased. (**C**,**D**) A 69-year-old male with neovascular glaucoma began to use a PG analog (bimatoprost) and exhibited increased vascularity.

**Table 1 jcm-09-00416-t001:** Demographic characteristics of the study population.

		Before Propensity Matching			After Propensity Matching		
		Aqueous Suppressants (*n* = 40)	Prostaglandin (PG) Analogs (*n* = 26)	*p*-Value	Aqueous Suppressants (*n* = 22)	PG Analogs (*n* = 22)	*p*-Value
Age (years)		56.0 ± 15.6	53.1 ± 17.2	0.479 ^a^	56.4 ± 14.3	55.2 ± 16.2	0.769 ^a^
Sex (male/female)		20/20	19/7	0.062 ^b^	11/11	17/5	0.116 ^b^
Laterality (Right/Left)		15/25	15/11	0.107 ^b^	9/13	13/9	0.366 ^b^
Diabetes (yes/no)		31/9	17/9	0.280 ^b^	6/16	9/13	0.526 ^b^
Hypertension (yes/no)		31/9	18/8	0.453 ^b^	6/16	8/14	0.747 ^b^
Axial Length (mm)		23.9 ± 8.4	25.3 ± 3.2	0.070 ^a^	23.9 ± 1.0	25.4 ± 3.4	0.098 ^a^
Central Corneal Thickness (μm)		538.5 ± 50.6	547.2 ± 64.4	0.584 ^a^	547.5 ± 53.6	558.8 ± 54.6	0.445 ^a^
Lens Status, no. (%)	Phakia	17 (42.5)	14 (53.8)	0.569 ^b^	9 (40.9)	11 (50)	0.451 ^b^
	Pseudophakia	22 (55)	11 (42.3)		13 (59.1)	10 (45.5)	
	Aphakia	1 (2.5)	1 (3.9)		0 (0)	1 (4.5)	
Glaucoma Subtype, no. (%)	POAG	14 (35)	6 (23.1)	0.799 ^b^	5 (22.7)	6 (27.3)	0.991 ^b^
	PACG	2 (5)	2 (7.7)		2 (9.1)	2 (9.1)	
	Uveitic glaucoma	10 (25)	5 (19.2)		6 (27.3)	5 (22.7)	
	Secondary OAG	3 (7.5)	4 (15.4)		3 (13.6)	3 (13.6)	
	Secondary ACG	6 (15)	5 (19.2)		2 (9.1)	3 (13.6)	
	NVG	5 (12.5)	4 (15.4)		4 (18.2)	3 (13.6)	
History of Intraocular Surgery, no. (%)	No surgery	11 (23.4)	7 (19.4)	0.232 ^b^	7 (31.8)	6 (27.3)	0.929 ^b^
	Cataract	20 (42.6)	12 (33.3)		7 (31.8)	6 (27.3)	
	Penetrating keratoplasty	3 (6.4)	5 (13.9)		1 (4.5)	2 (9.1)	
	Refractive surgery	0 (0)	2 (5.6)		0 (0)	0 (0)	
	Vitrectomy	5 (10.6)	2 (5.6)		3 (13.6)	2 (9.1)	
	Trabeculectomy	4 (8.5)	5 (13.9)		2 (9.1)	4 (18.2)	
	Ahmed implantation	4 (8.5)	1 (2.8)		2 (9.1)	2 (9.1)	
	Ex-PRESS implantation	0 (0)	2 (5.6)		0 (0)	0 (0)	
Visual Field (preoperative)	MD (dB)	−15.7 ± 11.7	−14.3 ± 13.9	0.674 ^a^	−15.5 ± 11.3	−14.9 ± 13.6	0.868 ^a^
	PSD (dB)	6.0 ± 3.8	4.2 ± 4.2	0.073 ^a^	5.9 ± 3.7	4.5 ± 4.2	0.178 ^a^
Duration of Follow-Up (months)		32.4 ± 8.3	31.4 ± 8.9	0.634 ^a^	33.8 ± 6.0	32.1 ± 8.4	0.451 ^a^

Mean values are presented with standard deviations ^a^ Student’s *t*-test, ^b^ chi-squared test. POAG, primary open-angle glaucoma; PACG, primary angle-closure glaucoma; OAG, open-angle glaucoma; ACG, angle-closure glaucoma; NVG, neovascular glaucoma; MD, mean deviation; PSD, pattern standard deviation.

**Table 2 jcm-09-00416-t002:** Postoperative complications.

	Before Propensity Matching			After Propensity Matching		
Yes/no (Percentage of Yes)	Aqueous Suppressants (*n* = 40)	PG Analogs (*n* = 26)	*p*-Value	Aqueous Suppressants (*n* = 22)	PG Analogs (*n* = 22)	*p*-Value ^b^
Choroidal Detachment	7/33 (17.5%)	6/20 (23.1%)	0.578 ^a^	4/18 (18.2%)	6/16 (27.3%)	0.721
Shallow Anterior Chamber	4/36 (10.0%)	6/20 (23.1%)	0.148 ^b^	3/19 (13.6%)	4/18 (18.2%)	1.000
Hyphemia	2/38 (5.0%)	5/21 (19.2%)	0.067 ^b^	1/21 (4.5%)	5/17 (22.7%)	0.185
Decompressive Retinopathy	0/40 (0.0%)	1/25 (3.8%)	0.211 ^b^	0/22 (0%)	1/21 (4.5%)	1.000
Disc Hemorrhage	1/39 (2.5%)	0/26 (0.0%)	0.417 ^b^	1/21 (4.5%)	0/22 (0%)	1.000
Hypotony	5/35 (12.5%)	5/21 (19.2%)	0.456 ^a^	5/17 (22.7%)	3/19 (13.6%)	0.698
Tube Malposition	3/37 (7.5%)	0/26 (0.0%)	0.273 ^b^	1/21 (4.5%)	0/22 (0%)	1.000
Tube Exposure	0/40 (0.0%)	0/26 (0.0%)	Not available	0/22 (0%)	0/22 (0%)	NA
Malignant Glaucoma	0/40 (0.0%)	0/26 (0.0%)	Not available	0/22 (0%)	0/22 (0%)	NA

Mean values are presented with standard deviations ^a^ Student’s *t*-test, ^b^ chi-squared test.

**Table 3 jcm-09-00416-t003:** Success rates assessed with two different criteria of the number of postoperative medications.

			Before Propensity Matching			After Propensity Matching		
Success Rates (%)		Time	Aqueous Suppressants (*n* = 40)	PG Analogs (*n* = 26)	*p*-Value	Aqueous Suppressants (*n* = 22)	PG Analogs (*n* = 22)	*p*-Value
**Complete Success**		1 year	7.5	0	0.273 ^a^	9.1	0	0.488 ^a^
		2 years	15.8	4.2	0.232 ^a^	14.3	4.8	0.606 ^a^
		3 years	13.9	5.0	0.405 ^a^	10.0	5.6	1.000 ^a^
**Qualified Success**	With the initial medication	1 year	67.5	42.3	**0.043**	68.2	50.0	0.220
		2 years	80.6	37.5	**0.001**	76.2	42.9	**0.028**
		3 years	80.0	35.0	**0.001**	80.0	38.9	**0.010**
	With 4 maximum medications	1 year	90	92.3	0.750 ^b^	95.5	90.9	1.000 ^b^
		2 years	88.9	91.7	0.725 ^b^	85.7	90.5	1.000 ^b^
		3 years	94.3	75.0	**0.039 ^b^**	95.0	72.2	0.083 ^b^

Complete success was defined as an IOP of 6 to 21 mmHg without any medications, while qualified success was defined as same IOP range but with up to four medications. The initial medication was one fixed-combination aqueous suppressant or one prostaglandin analogue. Bold font indicates significant *p* values (*p* < 0.05). ^a^ Fisher’s exact test; ^b^ chi-squared test.

**Table 4 jcm-09-00416-t004:** Logistic regression analysis of factors associated with surgical failure after Ahmed glaucoma valve (AGV) implantation.

	Before Propensity Matching				After Propensity Matching			
	Univariate		Multivariate		Univariate		Multivariate	
	Beta	*p*-Value	Beta	*p*-Value	Beta	*p*-Value	Beta	*p*-Value
Postoperative initial use of PG analogs	2.01	**0.001**	2.16	**0.002**	1.84	**0.013**	2.40	**0.009**
Previous glaucoma surgery	0.60	0.334			0.348	0.631		
Diabetes	−0.18	0.775			0.134	0.851		
Hypertension	−0.18	0.775			−0.383	0.600		
Axial length	0.19	0.204			0.121	0.419		
Maximum preoperative IOP	0.04	0.259			−0.001	0.990		
Number of preoperative total medication	0.76	0.077	0.77	0.161	0.906	0.140	1.33	0.139
Preoperative Visual field (MD)	0.04	0.111	0.04	0.167	0.037	0.175	0.033	0.313

IOP, intraocular pressure; PG, prostaglandin; MD, mean deviation. Bold font indicates significant *p* values (*p* < 0.05).

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
