# Peer review of "Initial Glaucoma Medication in the Hypertensive Phase Following Ahmed Valve Implantation: A Comparison of Results Achieved Using Aqueous Suppressants and Prostaglandin Analogs"

_jcm, 2020, doi:10.3390/jcm9020416_

Round 1

Reviewer 1 Report

Initial glaucoma medication in the hypertensive phase following Ahmed valve implantation: a comparison of results achieved using aqueous suppressants and prostaglandin analogs

There is a discrepancy between abstract and line 110 the in the manuscript where the authors state that "the qualified success was with a maximum of four medications" in the abstract and "two medications" on the experimental design section.

Section 3. Results has an initial paragraph including author instructions, which I find unnecessary for the readers.

Section 3.2 has a different font than the rest of the sections, the authors need to re-format.

Row 145-148 -it is not clear what the authors meant in terms of mean number of medications use statistical significance, and more specifically the word "however". Please clarify.

Author Response

Response to Reviewer 1 Comments

Point 1: There is a discrepancy between abstract and line 110 the in the manuscript where the authors state that "the qualified success was with a maximum of four medications" in the abstract and "two medications" on the experimental design section

Response 1: At the beginning of the study, we defined the qualified success with a maximum of two medications, however, we changed it with allowance of four medications. I, therefore, corrected the sentence at line 110 to four medications instead of two.

Point 2: Section 3. Results has an initial paragraph including author instructions, which I find unnecessary for the readers.

Response 2: I deleted the paragragh of author instructions.

Point 3: Section 3.2 has a different font than the rest of the sections, the authors need to re-format.

Response 3: I corrected the font of section 3.2 just like others. I appreciate your meticulous review.

Point 4: Row 145-148 -it is not clear what the authors meant in terms of mean number of medications use statistical significance, and more specifically the word "however". Please clarify.

Response 4: I rephase the sentence from line 145 to line 150 to clarity the mean number of medications use. Before prosensity score matching, the mean number of medications used was significantly greater in PG analogs group than AS group from postoperative 3 month to postoperative 3 years. After propensity score matching, the mean number of postoperative medications was significantly greater in PG analogs group than AS group at postoperative 3 months, 2 years and 3years. The difference in number of medications between groups was not statistically significant at postoperative 6 months and 1 year after propensity score matching.

Also, the word, “however”, was misused, therefore, I deleted it.

Reviewer 2 Report

How many patients were excluded who had follow-up of less than one year or who were followed up without medications?

2. Were the fixed-combination drops in the AS group (i.e. dorzolamide-timolol, brinzolamide-timolol) counted as one medication or two medications?

3. In the Discussion section, the authors state that the inferior outcomes due to PG analogs (when compared to AS group) was "due to the increased possibility of enhancing bleb fibrosis by PG-induced inflammation." Could there be other reasons such as: 1. Use of only one class of anti-glaucoma medications with PG group vs. two classes of medications in AS group; 2. PG group showed less IOP lowering efficacy since the Ahmed valve implant created new outflow channel that diminished ability of PG-mediated uveoscleral outflow pathway to function.

4. Did the authors use postoperative antifibrotics for modulation of postsurgical healing response (e.g. subconjunctival injections of 5-FU)? The reviewer has found this therapy very useful for reducing the need for initiating postoperative medication use in patients with the hypertensive phase following Ahmed valve implantation.

Author Response

Response to Reviewer 2 Comments

Point 1: How many patients were excluded who had follow-up of less than one year or who were followed up without medications?

Response 1: No one was excluded because of the short-term follow-up of less than one year.According to Section 3.1, 17 eyes who did not experience the hypertensive phase and were followed up without medications, were excluded. To clarify about exclusion criteria, I corrected the sentence of line 65 -66 which described the inclusion criteria and added the sentence at line 132 for more specification.        

Point 2: Were the fixed-combination drops in the AS group (i.e. dorzolamide-timolol, brinzolamide-timolol) counted as one medication or two medications?

Response 2: The fixed- combination drops in the AS group were counted as two medications. I made an additional comment about this at line 107-108.

Point 3: In the Discussion section, the authors state that the inferior outcomes due to PG analogs (when compared to AS group) was "due to the increased possibility of enhancing bleb fibrosis by PG-induced inflammation." Could there be other reasons such as: 1. Use of only one class of anti-glaucoma medications with PG group vs. two classes of medications in AS group; 2. PG group showed less IOP lowering efficacy since the Ahmed valve implant created new outflow channel that diminished ability of PG-mediated uveoscleral outflow pathway to function

Response 3: Majority of studies revealed that there are no differences in diminishing IOP between AS and PG analogs. I, however, did some more researches and found a few articles that the fixed combination of AS showed better IOP lowering potency than PG analogs in surgery naïve eyes(1, 2). Therefore, I guess your idea of that the greater the classes of medications patients use, the more the IOP drops could be reflected in our article. I made some changes of these from line 260 to line 262.

Also, PG -mediated outflow pathway might have been hindered due to the new channel created by the AGV implantation. I added your opinions at the end of discussion paragraphs (from line 268 to line 269).

Point 4: Did the authors use postoperative antifibrotics for modulation of postsurgical healing response (e.g. subconjunctival injections of 5-FU)? The reviewer has found this therapy very useful for reducing the need for initiating postoperative medication use in patients with the hypertensive phase following Ahmed valve implantation.

Response 4: There had been no postoperative interventions but resuming glaucoma medications. In our clinics, we have tried subconjunctival 5-FU injection after Ex-Press implantation or trabeculectomy, but we are willing to consider it for patients with hypertensive phase following AGV implantation.   Once we have some patients with postoperative 5-fu injections, it would be interesting to see and compare which postoperative intervention leads to most favourable short-term and long- term outcomes after AGV implantation.

Parmaksiz S, Yuksel N, Karabas VL, Ozkan B, Demirci G, and Caglar Y. A comparison of travoprost, latanoprost, and the fixed combination of dorzolamide and timolol in patients with pseudoexfoliation glaucoma. European journal of ophthalmology. 2006;16(1):73-80. Chiselita D, Antohi I, Medvichi R, and Danielescu C. Comparative analysis of the efficacy and safety of latanoprost, travoprost and the fixed combination timolol-dorzolamide; a prospective, randomized, masked, cross-over design study. Oftalmologia (Bucharest, Romania : 1990). 2005;49(3):39-45.

Round 2

Reviewer 2 Report

After reviewing the revised manuscript, the revised manuscript has been promoted